# LPCAT1 enhances castration resistant prostate cancer progression via increased mRNA synthesis and PAF production

Chao Han[1☯], Guopeng Yu[1☯], Yuanshen Mao[1☯], Shangqing Song[1], Long Li[1], Lin Zhou[1], Zhong Wang[1], Yushan Liu[1]*, Minglun Li[2]*, Bin Xu[1]*

**1** Department of Urology, Shanghai Ninth People's Hospital, Shanghai Jiaotong University School of Medicine, Shanghai, China, **2** Urologic and Hematologic Oncology, Department of Radiation Oncology, University Hospital, LMU Munich, Munich, Germany

☯ These authors contributed equally to this work.
* chxb2004@sina.com (BX); yushandoctor@163.com (YL); Minglun.li@med.uni-muenchen.de (ML)

**Data Availability Statement:** All relevant data are within the manuscript and its Supporting Information files.

## Abstract

Our previously study shown that Lysophosphatidylcholine Acyltransferase1 (LPCAT1) is overexpressed in castration resistant prostate cancer (CRPC) relative to primary prostate cancer (PCa), and androgen controls its expression via the Wnt signaling pathway. While highly expressed in CRPC, the role of LPCAT1 remains unclear. In vitro cell experiments referred to cell transfection, mutagenesis, proliferation, migration, invasion, cell cycle progression and apoptosis, Western blotting, Pulse-chase RNA labeling. BALB/c nude mice were used for in vivo experiments. We found that LPCAT1 overexpression enhanced the proliferation, migration, and invasion of CRPC cells both in vitro and in vivo. Silencing of LPCAT1 reduced the proliferation and the invasive capabilities of CRPC cells. Providing exogenous PAF to LPCAT1 knockdown cells increased their invasive capabilities; however platelet activating factor acetylhydrolase (PAF-AH) and the PAFR antagonist ABT-491 both reversed this phenotype; proliferation of CRPC cells was not affected in either model. LPCAT1 was found to mediate CRPC growth via nuclear re-localization and Histone H4 palmitoylation in an androgen-dependent fashion, increasing mRNA synthesis rates. We also found that LPCAT1 overexpression led to CRPC cell resistance to treatment with paclitaxel. LPCAT1 overexpression in CRPC cells drives tumor progression via increased mRNA synthesis and PAF production. Our results highlight LPCAT1 as a viable therapeutic target in the context of CRPC.

## Introduction

As an androgen-dependent disease, prostate cancer (PCa) is one of the most common genito-urinary tumors in men, and its incidence increases with age. In 2020, the United States is expected to have approximately 191,930 new cases and 33,330 deaths, surpassing lung cancer and becoming the most common malignant tumor in men [1]. To date, androgen deprivation therapy (ADT) is the gold standard regimen for advanced PCa patients, especially for those with metastasis [2]. While PCa progression is initially dependent on androgen, such ADT is

**Funding:** The authors acknowledge funding from the National Natural Science Foundation of China, 81472398, to Dr. Bin Xu; Shanghai Rising-Star Program, 15QA1404900, to Dr. Bin Xu; Interdisciplinary research of 9th People's Hospital affiliated to Shanghai Jiao Tong University School of Medicine, 201818042, to Dr. Bin Xu; and Fundamental Research Program Funding of Ninth People's Hospital affiliated to Shanghai Jiao Tong University School of Medicine, JYZZ006, to Dr. Guopeng Yu. The funders had no role in study design, data collection and analysis, decision to publish, or preparation of the manuscript.

**Competing interests:** The authors have declared that no competing interests exist.

**Abbreviations:** LPCAT1, Lysophosphatidylcholine Acyltransferase1; CRPC, castration resistant prostate cancer; PCa, prostate cancer; ADT, Androgen deprivation therapy; AIPC, androgen-independent prostate cancer; APC, androgen-dependent prostate cancer; PAF, platelet activating factor; DHT, dihydrotestosterone; PAF-AH, PAF-acetylhydrolases; FBS, fetal bovine serum; PI, propidium iodide.

often followed by a relapse owing to the growth of tumor cells in an androgen-independent manner [3]. This androgen-independent prostate cancer (AIPC), also known as castration-resistant prostate cancer (CRPC) is far deadlier than androgen-dependent prostate cancer (ADPC), and fewer clinical options exist for CRPC patients [4–6].

The exact mechanisms governing AIPC development are of great interest, but remain incompletely understood. We have previously reported on a key platelet activating factor (PAF) synthetase known as Lysophosphatidylcholine Acyltransferase1 (LPCAT1) that is highly expressed in CRPC tissue and cell samples, and we have shown that dihydrotestosterone (DHT) treatment induces Wnt/β-catenin-dependent PAF production and LPCAT1 expression [7]. Exactly what role LPCAT1 plays in CRPC progression, however, remains unclear. Zhou et al [8] have demonstrated that elevated LPCAT1 levels in PCa tissues allow it to be used as a prognostic biomarker of clinical outcomes following prostatectomy. LPCAT1 expression has also been shown to not be restricted to PCa, with high expression levels in and potential contributions towards the progression of breast cancer [9], clear cell renal cell cancer [10] and gastric cancer [11]. These findings suggest that LPCAT1 may have broad relevance in the context of cancer therapeutics.

PAF is a metabolite of arachidonic acid that acts as a pro-inflammatory phospholipid molecule [12]. Key roles for PAF have been identified in the context of processes including angiogenesis, proliferation, and migration. PAF-acetylhydrolases (PAF-AH) that degrade PAF into an inactive form via cleaving the sn-2 acetyl residue of PAF have also been identified [13]. We and others [7, 14–16] have demonstrated an oncogene-like role for PAF, wherein its signaling is linked with CRPC invasive potential, with elevated PAF levels in PCa samples being linked to LPCAT1 overexpression and the AA pathway [7]. These findings led us to hypothesize that LPCAT1 contributes to CRPC progression via promoting PAF release. To test this hypothesis, we performed siRNA-mediated knockdown of LPCAT1 in the C4-2 and PC-3 CRPC cell lines and assessed the proliferation and invasive capability of these cells. We further supplemented media with PAF in order to assess the relationship between PAF and any LPCAT1-dependent phenotype in these cells. We further validated these results via LPCAT1 overexpression and application of the PAFR antagonist ABT-491 as well as recombinant human PAH-AH was used to verify the effects of this overexpression. These experiments led us to last assess how LPCAT1 expression is associated with CRPC cellular responses to chemotherapeutic agents.

## Materials and methods

### Ethics statement

The institutional Ethical Committee of Shanghai Ninth Peoples' Hospital (Shanghai, China) approved this research (reference number: SH9H-2019-T181-2). Experimental animals were humanely treated based on the advice in the Guide for the Care and Use of Laboratory Animals.

### Experimental model development

In vivo overexpression experiments, 8-week-old male BALB/c nude mice (Slaccas Laboratory Inc., Shanghai, China) were used. A plasmid encoding for LPCAT1 or empty vector control was transfected into C4-2 cells. The 18 mice were randomly divided into overexpressing group (9 mice) and control group (9 mice). Following transfection, cells (overexpressing group and control) were suspended in a 50:50 mixture of PBS and Matrigel (BD, Franklin Lakes, NJ, USA), and $3 \times 10^6$ cells were subcutaneously injected into the flanks of these mice. Mice were monitored every 3 days and were sacrificed before the tumor volume reached 1000 mm$^3$. The tumor volume of all mice was less than 1000mm$^3$ in 6 weeks after transfection and no mice

died before meeting criteria for euthanasia. In order to alleviate suffering, all mice were sacrificed by cervical spine dislocation after anesthesia by Isoflurane.

To assess whether LPCAT1 overexpression can promote tumor growth, 6 mice (3 overexpressing and 3 control) were euthanized and tumors were harvested and weighed in 6 weeks after transfection.

To assess how androgen influenced tumor growth in the presence or absence of LPCAT1 overexpression, 6 mice (3 overexpressing and 3 control) 's murine testes were removed 2 weeks following tumor cell implant. After 6 weeks, 6 mice were euthanized and tumors were harvested and weighed.

To assess cell sensitivity to chemotherapeutic agents upon LPCAT1 overexpression, paclitaxel (20mg/kg) or DMSO vehicle control was injected intra-peritoneally into 6 mice (3 overexpressing and 3 control)(i.p.) daily for 5 days, starting 14 days after tumor cell implant as previously described [14]. After a total of 6 weeks, 6 mice were sacrificed as above, with tumor volumes being measured every two days.

## Cell culture

C4-2 and PC-3 cells (Stem Cell Bank, Chinese Academy of Sciences, Shanghai, China), were grown in RPMI-1640 containing 10% fetal bovine serum (FBS) (Gemini, 100–700) or Dextran Stripped FBS (Gemini, 100–119) as well as 1% penicillin/streptomycin (P/S). All cell lines were maintained at 37°C in a humidified incubator (5% $CO_2$) and routinely tested and authenticated using a panel of genetic and epigenetic markers by STR.

## Cell transfection

For cell interference, $2 \times 10^5$ cells were added to a 6 well plate, and 12 hours later LPCAT1 siRNA (SR312680; Origene Biotech, Beijing, China) or luciferase siRNA (control) was transfected into these cells via Roche reagent (04476093001,Basel, Switzerland) according to the manufacturer's protocol. After 48 hours, transfection efficiency was assessed. To overexpress LPCAT1, full-length human LPCAT1 cDNA (a gift from Han Lab) was cloned into the pCDH-CMV-MCS-EF1-puro vector. Empty vector was control group. This or a control plasmid was then cotransfected with pMD.2G and PSPAX2 into HEK-293FT cells with Lipofectamine LTX (Invitrogen, Carlsbad, CA, USA) according to the manufacturer's protocol. Culture supernatant was harvested after 48 hours, and the viruses therein were used to infect C4-2 cells twice at 24 hour intervals. Puromycin was then used to select for transduced cells.

## Mutagenesis

GenePharma Co., Ltd (Suzhou, China) designed and constructed the Histone H4 S47A mutation, as previously described.

## Cell proliferation, migration and invasion measurements

Cell proliferation was evaluated via MTS assay (RS Biotechnology,Shanghai), while cell mobility was evaluated by transwell assay as previously described [7].

## Cell cycle progression and apoptosis

To measure cell cycle progression, cells were fixed with 70% ethanol at 4°C for 24 h, stained with with propidium iodide (PI) (Abcam, USA) with RNase A for 30 min in PBS, and were then analyzed via flow cytometry. To assess apoptosis, cells were suspended in 100 μl 1 x

Annexin V binding buffer, and stained with Annexin V and PI (BD Pharmingen; 5 μl/tube) for 15 minutes at room temperature prior to flow cytometry analysis.

## Western blotting

Lysis buffer was used to collect total cell protein for 30 min at 4˚C, while a Nuclear and Cytoplasmic Protein Extraction Kit (Beyotime, China, P0027) was employed for nuclear protein extraction based on the provided instructions. The Bradford kit (Beyotime, China) was used to quantify all protein samples, which were then run on an SDS-PAGE gel and transferred to PVDF membranes. 5% non-fat milk in TBST was used to block membranes for 1 hour at room temperature, and then primary antibodies (LPCAT1, Abcam, ab214034; Histone H3.1, Cell Signaling Technology, 9717; active RNA polymerase II, Millipore, 04–1570; β-actin, Abcam, ab8227) were used to probe blots at 4˚C overnight. Blots were then washed three times and incubated with an HRP-conjugated secondary antibody for 1 h prior to visualization with Super Signal West Pico Chemiluminescent Substrate (Thermo Fisher Scientific, Massachusetts, MA, USA).

## Pulse-chase RNA labeling

Cells in exponential growth were pulse-chase-labeled with 1μCi of [$^{32}$P] UTP for 2 h. TRIzol was used to isolate total RNA, and scintillation counting was used to determine radioactive incorporation.

## PAF concentration measurement

PAF levels in supernatant samples were assessed via ELISA Kit (F02220, Westang Biotech, China) based on provided instructions. When assessing the effects of human recombinant Platelet Activating Factor Acetylhydrolase (PAH-AH) (PeproTech EC Ltd., USA) treatment, supernatants were collected 24 hours following the addition of PAH-AH.

## Statistical analysis

Results are given as means ± s.e.m. Differences between groups were assessed via one-way ANOVA and Student-Newman-Keuls test. The significance threshold was $P<0.05$.

# Results

## LPCAT1 knockdown alters CRPC progression

To explore how LPCAT1 expression influences CRPC progression, we used LPCAT1 siRNA to knockdown LPCAT1 in the C4-2 and PC-3 CRPC cell lines and control group was luciferase siRNA (Fig 1A). Silencing LPCAT1 markedly reduced the proliferation of both of these cell lines after 48 and 72 hours (Fig 1B), and also decreased the migration and invasion of these cells (Fig 1C and 1D). In order to assess whether this was linked with a downregulation of PAF, we measured PAF levels and found them to be decreased upon LPCAT1 silencing (Fig 1E). We next provided exogenous PAF to cells that had received the LPCAT1 siRNA and the luciferase siRNA, and we observed no effect of PAF on cell proliferation regardless of dose ($10^{-8}$–$10^{-6}$ M; Fig 1F). This exogenous PAF did, however, facilitate a dose-dependent increase in the motility of cells that had been reduced upon LPCAT1 knockdown, and the PAF receptor antagonist ABT-491 disrupted this effect (Fig 1G and 1H). Together, these results indicate that knocking down LPCAT1 can impair the growth and invasive capabilities of CRPC cells via the PAF-PAFR pathway.

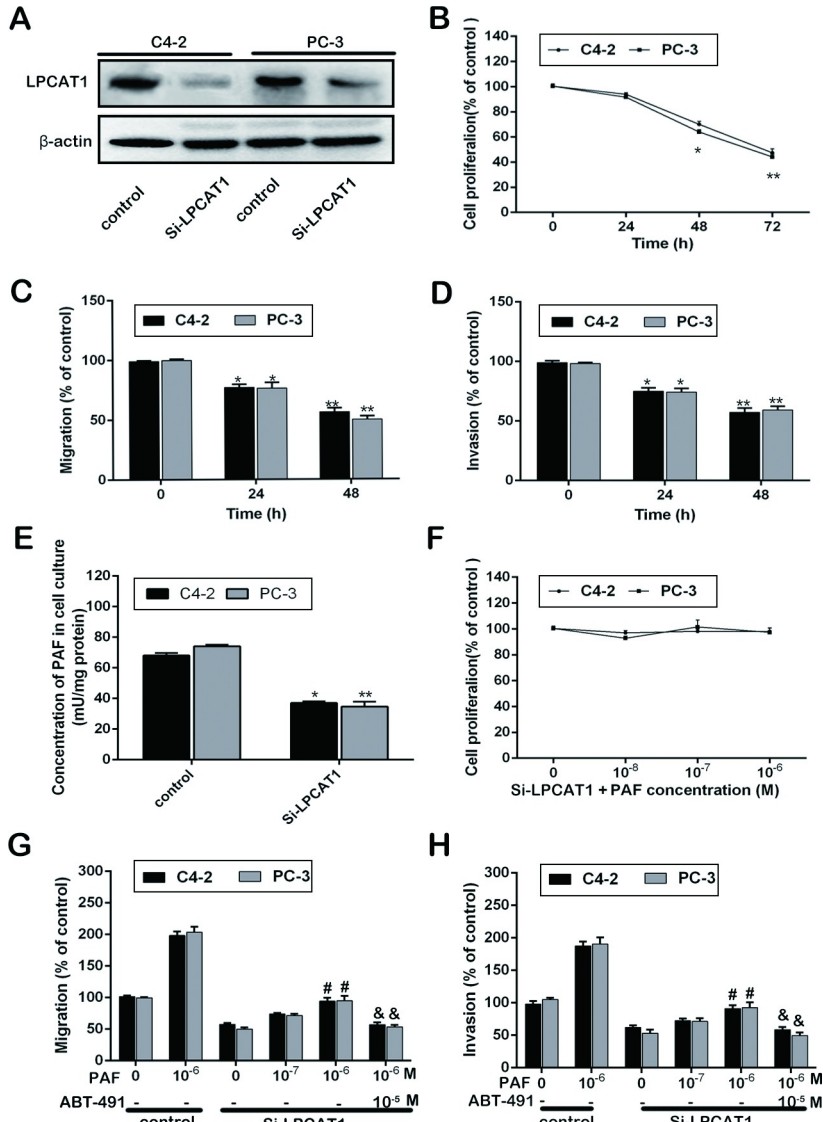

**Fig 1. LPCAT1 knockdown alters CRPC progression.** (A): LPCAT1 Si-RNA was transfected into C4-2 and PC-3 cells, and transfection efficiency was demonstrated. (B): The ability of cell proliferation was measured at 0, 24, 48 and 72 h after transfection. (C) and (D): The ability of cell migration and invasion were determined at 0, 24 and 48h after transfection. (E): The PAF concentration in cell cultures of Si-LPCAT1 group and control at 48 h after transfection. (F): Cell proliferation assay were measured at 48 h after C4-2 and PC-3 cells in Si-LPCAT1 group were treated with increasing concentration of PAF ($10^{-8}$–$10^{-6}$ M) and the control group. (G) and (H): Cell migration and invasion assay were measured at 48 h after increasing concentration of PAF ($10^{-8}$–$10^{-6}$ M) in Si-LPCAT1 group and control group, and ABT-491 ($10^{-5}$ M) was used to pretreat cells induced by PAF ($10^{-6}$ M). Values are presented as mean percent control ± s.e.m or mean ± s.e.m. * $P < 0.05$ and ** $P < 0.01$ compared with control; # $P < 0.05$ and ## $P < 0.01$ compared with Si-LPCAT1 treated with PAF (0 M); & $P < 0.05$ and && $P < 0.01$ compared with Si-LPCAT1 treated with PAF ($10^{-6}$ M).

## LPCAT1 overexpression alters motility in CRPC cells

LPCAT1 vector and empty vector (control) was transfected into C4-2 cells, and the transfection efficiency in protein level was demonstrated via WB. And the result showed that LPCAT1 protein level in LPCAT1 group (LPCAT1-overexpression cells) had a higher expression compared to control group (Fig 2A). LPCAT1 overexpression was associated with enhanced

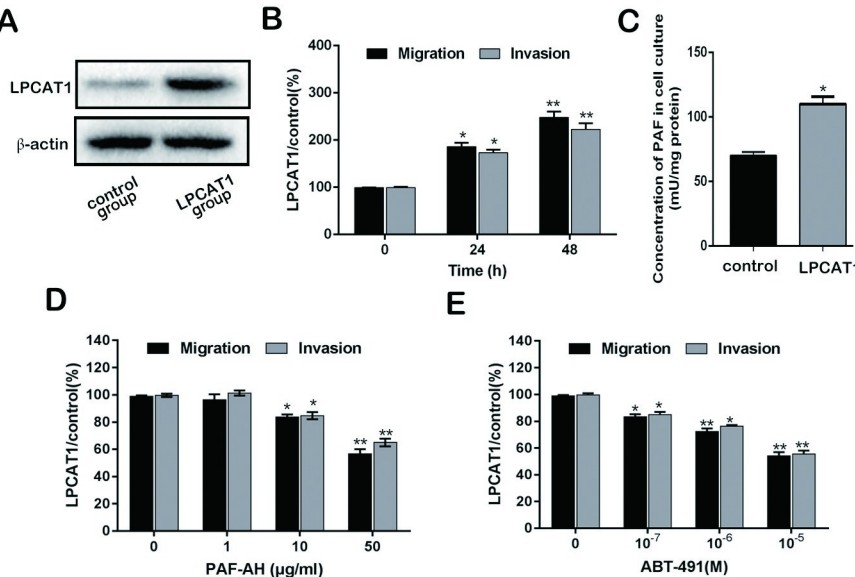

**Fig 2. LPCAT1 overexpression alters motility in CRPC cells.** (A): LPCAT1 vector and empty vector (control) was transfected into C4-2 cells, and the transfection efficiency was demonstrated. (B): the ability of cell migration and invasion was determined at 0, 24 and 48 h after transfection. (C): The PAF concentration in cell cultures of LPCAT1 group and control group. (D) and (E): Cell migration and invasion assay were measured at 48 h after increasing concentration of PAF-AH (1–50 µg/ml) and ABT-491($10^{-7}$–$10^{-5}$ M) in LPCAT1 group and control group. Values are presented as mean percent control ± s.e.m or mean ± s.e.m. $^*$ P < 0.05 and $^{**}$ P < 0.01 compared with control.

migration and invasion relative to controls (Fig 2B). PAF levels were also increased in the context of LPCAT1 overexpression (Fig 2C). In order to determine whether LPCAT1 was linked to alter PAF release, we provided cells with PAF-AH or ABT-491 to degrade PAF or inhibit PAFR, respectively. Migration/invasion were assayed at 48h. PAF-AH treatment suppressed the LPCAT1 overexpression-enhanced migration and invasion (Fig 2D), as did ABT-491 (Fig 2E) in a dose-dependent fashion. Together these findings further support a role for LPCAT1 in mediate the migratory and invasive capabilities of CRPC cells via PAF-PAFR signaling.

## LPCAT1 overexpression alters CRPC cell growth

LPCAT1 group (LPCAT1-overexpressing C4-2 cells) exhibited enhanced proliferation relative to controls (Fig 3A), with a colony formation assay yielding similar results (Fig 3B). LPCAT1 overexpression was also associated with increased frequencies of cells in the G2 cell cycle phase as well as decreased frequencies of those in the G1/S phases, as measured by a cell cycle assay (Fig 3C). To extend these findings to a more physiologically relevant setting, we conducted a xenograft tumor experiment to assess LPCAT1-dependent tumor growth. This experiment demonstrated increased tumor growth in cells expressed elevated levels of LPCAT1 (p < 0.05; Fig 3D). Given the above findings of elevated PAF levels upon LPCAT1 overexpression, we next provided PAF-AH to cultured cells to degrade exogenous PAF. While PAF was degraded successfully, no differences in proliferation were observed upon such treatment, indicating that the LPCAT1-dependent enhancement of cell proliferation is independent of extracellular PAF (Fig 3E and 3F).

## Androgen promotes LPCAT1 nuclear relocalization

Our results thus far suggest that in CRPC cells LPCAT1 promotes migration in a PAF-dependent fashion, whereas it promotes proliferation through PAF-independent mechanisms.

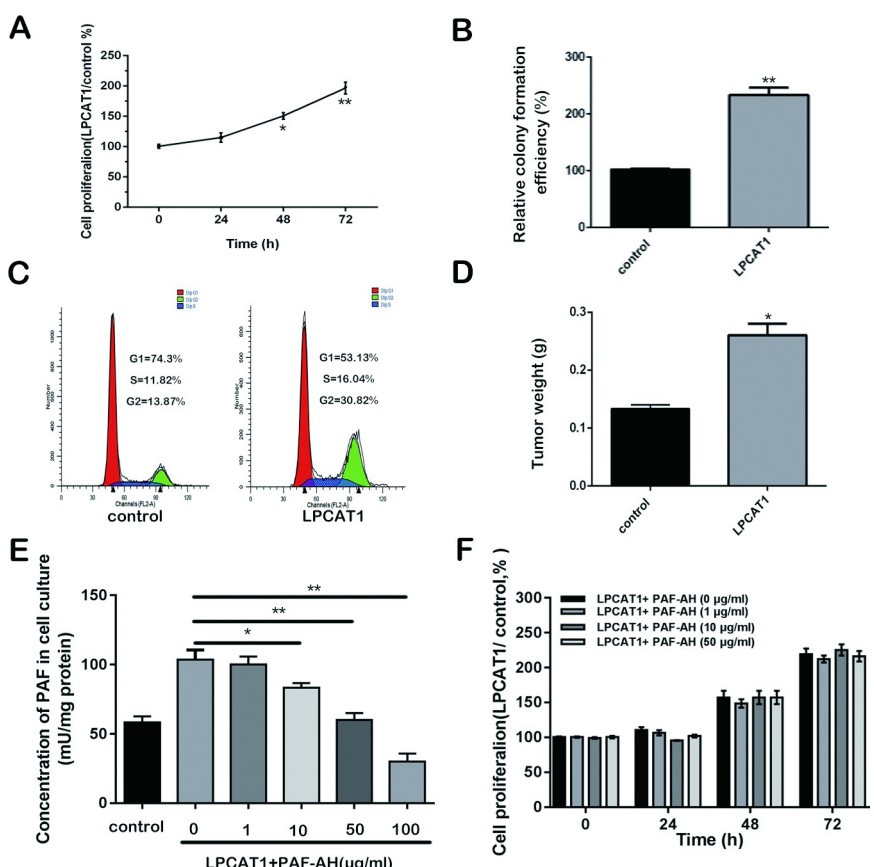

**Fig 3. LPCAT1 overexpression alters CRPC cell growth.** (A): The ability of Cell proliferation was measured at 0, 24, 48 and 72 h after transfection. (B): Colony formation efficiency was measured in LPCAT1 group and control group. (C): cell cycle was measured in LPCAT1 group and control group. (D): Tumorigenicity assay in 6 nude mice was performed and tumors weights in LPCAT1 and control groups were measured. (E): The PAF concentration was measured at 48 h after PAF-AH treatment. (F): The ability of cell proliferation in LCPAT1 group with different concentration PAF-AH treatment and control group was measured at 0, 24, 48 and 72h. Values are presented as mean percent control ± s.e.m or mean ± s.e.m. * P < 0.05 and ** P < 0.01 compared with control.

Previous work has shown that LPCAT1 can localize to the nucleus wherein it can promote enhanced mRNA production [15]. To test the relevance of this finding to our phenotype, and to determine whether androgen might mediate such nuclear relocalization, we measured nuclear LPCAT1 in C4-2 cells exposed to various androgen doses, along with increasing DHT doses to test the androgen-dependence of observed phenotypes. It was found that LPCAT1 expression in nucleus was increased in C4-2 cells transfected with LPCAT1 compared the control cells cultured in RPMI-1640 medium containing a certain amount of androgen (Fig 4A). We postulated that androgen may participate in the process of LPCAT1 nucleus shift. Therefore, cells were treated with increasing concentration of DHT. As predicted, androgen increased LPCAT1 nuclear localization in a dose dependent fashion (Fig 4B). Next, LPCAT1-overexpressing C4-2 cells and control were simultaneously cultured with DHT ($10^{-6}$ M), DHT ($10^{-6}$ M) + Flu (flutamide) or DHT (0 M). When cells overexpressing LPCAT1 were treated with DHT ($10^{-6}$ M), proliferation was increased, and the effect was reversed by Flu (androgen receptor antagonist flutamide) (Fig 4C). Moreover, tumorigenesis assay showed that there was no significant difference in tumor weight between LPCAT1 overexpressing group and control when male nude mice was castrated at 6 weeks (Fig 4D). These findings

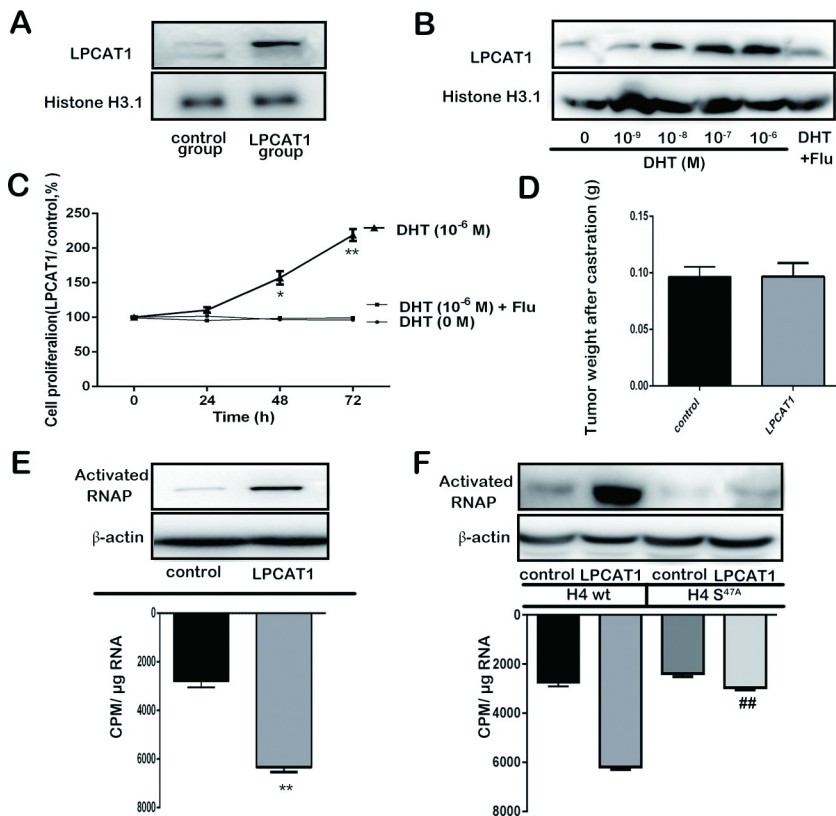

**Fig 4. Androgen promotes LPCAT1 nuclear relocalization.** (A): Intra-nuclear LPCAT1 expression in LPCAT1 and control groups and cells were cultured with ordinary FBS containing a certain amount of androgen. (B): The effect of the increased concentrations of DHT on intra-nuclear LPCAT1 expression of C4-2 cells cultured in Dextran Stripped FBS was detected. (C): The effect of DHT ($10^{-6}$ M), DHT ($10^{-6}$ M) +Flu and DHT (0 M) on the proliferation ability of C4-2 cells cultured in Dextran Stripped FBS was detected. (D): Tumors weights in 6 nude mice with LPCAT1 and control groups after castration. (E): Activated RNA polymerase (RNAP) Ⅱ expression (up) and total RNA synthesis (down) were measured in LPCAT1 and control groups cells with ordinary FBS culture. (F): Histone H4 S47A mutation was designed and transfected into LPCAT1 and control groups, and activated RNA polymerase (RNAP) Ⅱ expression (up) and total RNA synthesis (down) were measured in WT and mutation groups. Values are presented as mean percent control ± s.e.m or mean ± s.e.m. * $P < 0.05$ and ** $P < 0.01$ compared with control. # $P < 0.05$ and ## $P < 0.01$ compared with LPCAT1 overexpressing cells in WT group.

suggest that androgen induced LPCAT1 nuclear localization in CRPC cells, thereby driving proliferation. We next measured RNA synthesis and polymerase (RNAP) Ⅱ activation upon LPCAT1 overexpression in the presence of varying androgen doses, and we found both of these activities to be increased upon LPCAT1 overexpression (Fig 4E). O-palmitoylation of histone H4 has been shown to mediate this increase in mRNA synthesis [15]. To test whether this was the case in our model, we constructed a histone H4 S47A mutation which was transfected into cells, leading to decreases in LPCAT1-dependent RNA synthesis and RNAPⅡacti-vation (Fig 4F). This indicates a role for O-palmitoylation of histone H4 as a mediator of LPCAT1-induced increases in mRNA synthesis.

## LPCAT1 expression regulates paclitaxel sensitivity

LPCAT1 has been shown to make lung cancer cells more sensitive to cisplatin, but how it alters the chemosensitivity of CRPC cells is unknown. We therefore gave paclitaxel to C4-2 cells overexpressing LPCAT1 or controls, and found that elevated LPCAT1 levels reduced CRPC

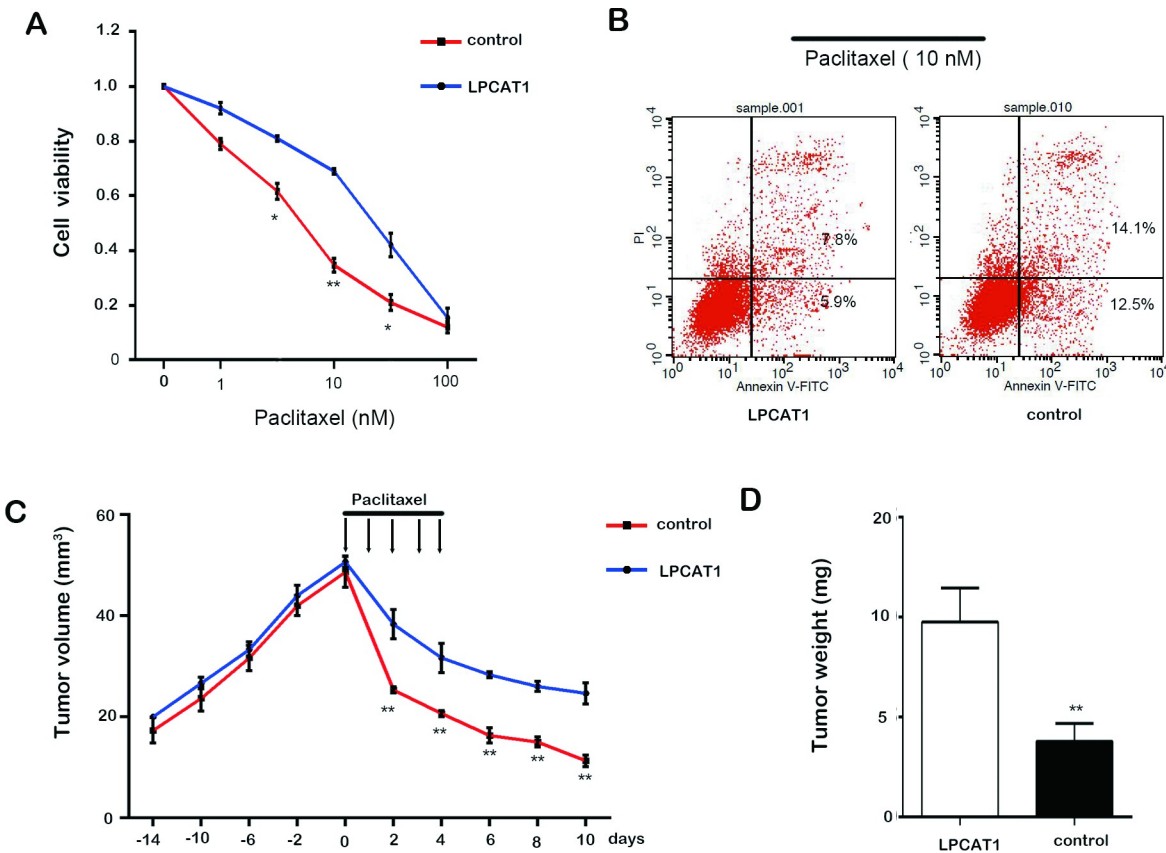

**Fig 5. LPCAT1 expression regulates paclitaxel sensitivity.** (A): Cell viability was determined in LPCAT1 and control groups with the increasing concentration of paclitaxel treatment (1, 10 and 100 nM). (B): Cell apoptosis in LPCAT1 and control groups with paclitaxel (10 nM) treatment. (C) and (D): Cells were injected into 6 nude mice to initiate tumorigenesis, with paclitaxel treatment administered 14 days post inoculation. Tumor volumes at different time points (C) and tumors weights (D) at excision. Values are presented as mean ± s.e.m. * P < 0.05 and ** P < 0.01 compared with control.

cell sensitivity to paclitaxel (Fig 5A). LPCAT1 overexpression similarly led to reduced paclitaxel-induced apoptosis relative to control cells (P<0.05) (Fig 5B). To test this phenotype in vivo, mice given LPCAT1-overexpressing or control tumors were treated with or without paclitaxel beginning 3 weeks after tumor implantation. Prior to paclitaxel dosing, LPCAT1 led to little increase in tumor growth (Fig 5C), whereas upon paclitaxel treatment growth was more completely suppressed in control tumors than in LPCAT1 overexpressing tumors (Fig 5C). This was consistent with differences in tumor weight (Fig 5D), and suggests that LPCAT1 renders CRPC cells resistant to paclitaxel therapy in vivo.

## Discussion

In this study, we investigated the molecular mechanisms by which LPCAT1 contributes to the progression of CRPC. We showed that LPCAT1 drives the proliferation and invasive potential of CRPC cells both in vitro and in vivo. LPCAT1-induced increases in cell migration and invasion were associated with PAF signaling, such that PAF-AH ablated this enhanced motility upon LPCAT1 overexpression, whereas exogenous PAF restored this motility upon LPCAT1 knockdown. LPCAT1-mediated increases in cell proliferation were unrelated to PAF expression, and were instead linked to androgen-dependent LPCAT1 nuclear localization, Histone

H4 O-palmitoylation, and increased synthesis of mRNA. Overexpressing LPCAT1 also made CRPC cells less sensitive to paclitaxel.

While in many cases local PCa can be readily and effectively treated, upon progression to CRPC this disease remains extremely deadly [16]. Indeed, PCa is still the second most common source of cancer-related death in men in many countries [17], making it vital that novel therapeutic treatment avenues be identified. LPCAT1 can enhance the proliferation and invasion of CRPC cells, while PAF is involved in the progression of many cancers via PAFR binding and signaling activation [18]. PAF can be synthesized via a remodeling pathway in which LPCAT1 converts lyso-PAF into PAF. Another lyso-PAF acetyltransferase, LPACT2, can also produce PAF from lyso-PAF and acetyl-CoA [19]. LPCAT2 expression is associated with more aggressive PCa according to genetic analyses [20]. Whether it is linked with PAF production in the context of PCa, however, warrants further investigation. Some groups have identified a role for PAF as a promoter of breast, esophageal, and ovarian cancer progression through PAFR signaling pathways [21], we have previously [7] found PAF to be involved in the migration but not the proliferation of C4-2 cells.

LPCAT1 nuclear localization has been shown to occur in response to a range of stimuli, including bacterial and LPS stimulation, as well as changes in $Ca^{2+}$ stimulation [15, 22]. Once in the nucleus, LPCAT1 can drive the palmitoylation of histone H4, ultimately enhancing mRNA synthesis and making such palmitoylation a novel marker of inflammation [22]. Given these previous observations, we predicted that LPCAT1 may play a similar role upon androgen stimulation in the context of CRPC. Consistent with this hypothesis, we observed increased nuclear LPCAT1 following androgen treatment both in vitro and in castrated mice. Increased $Ca^{2+}$ flux upon DHT treatment [23] may also contribute to LPCAT1 nuclear re-localization and subsequent increased mRNA synthesis.

Palmitoylation is a common form of post-translational modification. While S-palmitoylation is fairly well studied, O-palmitoylation is less well understood. One study has linked such O-palmitoylation to the regulation of Wnt protein secretion and subsequent Wnt signaling pathway inactivation [24]. We observed increased histone H4 O-palmitoylation upon LPCAT1 overexpression, and this was associated with increased mRNA synthesis. Histones themselves can play key roles in a wide range of diseases depending on their specific post-translational modifications, which alter their functionality [25]. Alterations in histone phosphorylation, acetylation, or methylation have been linked to CRPC progression owing to the epigenetic modifications that result therefrom [26]. Histone deacetylase inhibitors can disrupt CRPC progression and have been tested in clinical settings for this reason [27, 28]. Histone H4 O-palmitoylation may be a useful therapeutic target or biomarker of CRPC progression, and further study is thus warranted.

In Conclusion, LPCAT1 overexpression in CRPC cells drives tumor progression, both via PAF-mediated enhancement of tumor motility and LPCAT1 nuclear-localization-mediated histone H4 O-palmitoylation in an androgen-dependent but AR-independent fashion, ultimately enhancing mRNA synthesis and cell proliferation.

## Supporting information

**S1 Original images.**
(PDF)

**S1 File.**
(DOCX)

## Acknowledgments

We are very grateful to Han Lab for presenting the full-length cDNA of LPCAT1 to us and based on this, we can carry out follow-up experiments to study the role of LPCAT1 in the progression of CRPC.

## Author Contributions

**Conceptualization:** Yushan Liu, Minglun Li, Bin Xu.

**Formal analysis:** Chao Han, Guopeng Yu, Yuanshen Mao, Long Li.

**Funding acquisition:** Guopeng Yu, Bin Xu.

**Investigation:** Chao Han.

**Methodology:** Chao Han, Guopeng Yu, Yuanshen Mao, Shangqing Song, Long Li, Lin Zhou.

**Writing – original draft:** Chao Han, Guopeng Yu, Yuanshen Mao, Zhong Wang, Yushan Liu, Minglun Li, Bin Xu.

**Writing – review & editing:** Chao Han, Guopeng Yu, Yuanshen Mao, Zhong Wang, Yushan Liu, Minglun Li, Bin Xu.

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
