## [Decision Letter · Decision Letter 0]

18 Aug 2020

PONE-D-20-13714

LPCAT1 enhances castration resistant prostate cancer progression via increased mRNA synthesis and PAF production

PLOS ONE

Dear Dr. Xu,

Thank you for submitting your manuscript to PLOS ONE. After careful consideration, we feel that it has merit but does not fully meet PLOS ONE’s publication criteria as it currently stands. Therefore, we invite you to submit a revised version of the manuscript that addresses the points raised during the review process.

Please revise the manuscript according to the suggestions made by the reviewer. 

We look forward to receiving your revised manuscript.

Kind regards,

Chih-Pin Chuu, Ph.D.

Academic Editor

PLOS ONE

Journal Requirements:

2. To comply with PLOS ONE submissions requirements, in your Methods section, please provide additional information on the animal research and ensure you have included details on (1) methods of sacrifice, (2) methods of anesthesia and/or analgesia, and (3) efforts to alleviate suffering.

3. As part of your revision, please complete and submit a copy of the ARRIVE Guidelines checklist, a document that aims to improve experimental reporting and reproducibility of animal studies for purposes of post-publication data analysis and reproducibility: https://www.nc3rs.org.uk/arrive-guidelines. Please include your completed checklist as a Supporting Information file. Note that if your paper is accepted for publication, this checklist will be published as part of your article.

4.PLOS ONE now requires that authors provide the original uncropped and unadjusted images underlying all blot or gel results reported in a submission’s figures or Supporting Information files. This policy and the journal’s other requirements for blot/gel reporting and figure preparation are described in detail at https://journals.plos.org/plosone/s/figures#loc-blot-and-gel-reporting-requirements and https://journals.plos.org/plosone/s/figures#loc-preparing-figures-from-image-files. When you submit your revised manuscript, please ensure that your figures adhere fully to these guidelines and provide the original underlying images for all blot or gel data reported in your submission. See the following link for instructions on providing the original image data: https://journals.plos.org/plosone/s/figures#loc-original-images-for-blots-and-gels.

Reviewers' comments:

Reviewer's Responses to Questions

**Comments to the Author**

1. Is the manuscript technically sound, and do the data support the conclusions?

Reviewer #1: Yes

2. Has the statistical analysis been performed appropriately and rigorously? 

Reviewer #1: Yes

3. Have the authors made all data underlying the findings in their manuscript fully available?

Reviewer #1: Yes

4. Is the manuscript presented in an intelligible fashion and written in standard English?

Reviewer #1: Yes

5. Review Comments to the Author

Reviewer #1: Prostate cancer is a very common malignant tumor, and there are many cases with either recurrence after primary local therapy or with advanced stage at first diagnosis. First-line therapy in this situation is usually androgen deprivation. After some time, there is often development of a castration-resistant prostate cancer (CRPC), and treatment options at this stage are still limited, because knowledge about mechanisms driving CRPC is limited. There is published evidence that LPCAT1 is highly expressed in CRPC tissue, and that it is a prognostic biomarker in prostate cancer. Furthermore, various papers demonstrate an oncogenic function of PAF in different tumors.

The aim of this study was to analyze the role of LPCAT1 and PAF in CRPC in further detail. The authors performed different cell culture experiments on two prostate cancer cell lines, and also some in vivo experiments on nude mice. They found that LPCAT1 can enhance tumor cell migration and invasion in a PAF-dependent manner. In contrast, cell proliferation was independent of PAF, and instead associated with androgen-dependent nuclear localization of LPCAT1. In the nucleus, LPCAT1 can drive palmitoylation of histone H4, leading to increased mRNA synthesis. Furthermore, elevated LPCAT1 levels reduced the sensitivity of prostate cancer cells to paclitaxel, which is a common drug in CRPC.

The topic of this study has high relevance in the field of advanced prostate cancer, because only a few mechanisms driving CRPC progression are known so far, and new potential therapy targets are urgently needed. Therapeutic modulation of LPCAT1, PAF or epigenetic modifications like histone palmitoylation might be novel approaches in the future. The strength of this study is the comprehensive experimental approach including many different cell culture experiments (measurement of cell cycle, mRNA level, proliferation, migration, invasion, apoptosis, gene transfection and silencing, Western Blot) and in vivo experiments with tumor xenografts in nude mice. The experiments are well planned, and the results give new insights in cell biology mechanisms that were previously not known in CRPC. The presentation of the results in the text and in the figures is concise and clear, the discussion is concludent.

Minor revision:

In 3.4. (Results) a histone H4 S47A mutation was transfected into the tumor cells, resulting in a decreased LPCAT1-dependent mRNA synthesis and reduced RNAP II activation (Figure 4F). It is concluded that this indicates a role for O-palmitoylation of histone H4. The authors should explain in some more detail the effect of this mutation and the conclusion of O-palmitoylation as a potential mechanism in CRPC biology.

Summary:

This well conducted study gives new insight in cell biology mechanisms that play are role in CRPC. LPCAT1 might be a target for therapeutic approaches in the future (e.g., modulation of LPCAT1 to increase sensitivity against paclitaxel), albeit this basic research results can be only the very first step on the way to clinical application.

6. PLOS authors have the option to publish the peer review history of their article (what does this mean?). If published, this will include your full peer review and any attached files.

Reviewer #1: No

---

## [Author Response · Author response to Decision Letter 0]

18 Sep 2020

1、I reformatted and revised the manuscript to meet PLOS ONE's style requirements.

2、In my Methods section, I added details on (1) methods of sacrifice in , (2) methods of anesthesia and/or analgesia, and (3) efforts to alleviate suffering

in the last sentence of the second paragraph，which is“In order to alleviate suffering, all mice were sacrificed by cervical spine dislocation after anesthesia by Isoflurane”. And in order to alleviate suffering, we also designed to euthanize the mice before the tumor volume reached 1000 mm3.

3、I completed and submitted a copy of the ARRIVE Guidelines checklist.

4、The original pictures have been provided, but they had been edited. We regret that the original uncropped and unadjusted images were not saved at that time. But fortunately we still have samples, we have repeated the experiments and supplied the original uncropped and unadjusted images of Western Blotting. The results of the two experiments are exactly the same, and the conclusions drawn are also the same. We guarantee that our experiments are completely repeatable.

5、I am sure that I have an ORCID ID and that it is validated in Editorial Manager.

6、To reviewer：A histone H4 S47A mutation was transfected into the tumor cells, resulting in a decreased LPCAT1-dependent mRNA synthesis and reduced RNAP II activation (Fig 4F). This indicates a role for O-palmitoylation of histone H4 as a mediator of LPCAT1-induced increases in mRNA synthesis. O-palmitoylation of histone H4 has been shown to mediate this increase in mRNA synthesis. So increased LPCAT1 mRNA synthesis will enhance castration resistant prostate cancer progression.

---

## [Decision Letter · Decision Letter 1]

5 Oct 2020

LPCAT1 enhances castration resistant prostate cancer progression via increased mRNA synthesis and PAF production

PONE-D-20-13714R1

Dear Dr. Xu,

We’re pleased to inform you that your manuscript has been judged scientifically suitable for publication and will be formally accepted for publication once it meets all outstanding technical requirements.

Kind regards,

Chih-Pin Chuu, Ph.D.

Academic Editor

PLOS ONE

Additional Editor Comments (optional):

Reviewers' comments:

Reviewer's Responses to Questions

**Comments to the Author**

1. If the authors have adequately addressed your comments raised in a previous round of review and you feel that this manuscript is now acceptable for publication, you may indicate that here to bypass the “Comments to the Author” section, enter your conflict of interest statement in the “Confidential to Editor” section, and submit your "Accept" recommendation.

Reviewer #1: All comments have been addressed

2. Is the manuscript technically sound, and do the data support the conclusions?

Reviewer #1: Yes

3. Has the statistical analysis been performed appropriately and rigorously? 

Reviewer #1: Yes

4. Have the authors made all data underlying the findings in their manuscript fully available?

Reviewer #1: Yes

5. Is the manuscript presented in an intelligible fashion and written in standard English?

Reviewer #1: Yes

6. Review Comments to the Author

Reviewer #1: (No Response)

7. PLOS authors have the option to publish the peer review history of their article (what does this mean?). If published, this will include your full peer review and any attached files.

Reviewer #1: No

---

## [Editor Report · Acceptance letter]

23 Oct 2020

PONE-D-20-13714R1 

LPCAT1 enhances castration resistant prostate cancer progression via increased mRNA synthesis and PAF production 

Dear Dr. Xu:

I'm pleased to inform you that your manuscript has been deemed suitable for publication in PLOS ONE. Congratulations! Your manuscript is now with our production department. 

Kind regards, 

on behalf of

Prof. Chih-Pin Chuu 

Academic Editor

PLOS ONE